# Factors influencing users' willingness to use new energy vehicles

**Jianjun Pang**[1]*, **Jing Ye**[2], **Xuan Zhang**[3]

**1** Department of Industrial Design, College of Design, Jiaxing University, Jiaxing, China, **2** Department of Fashion Design and Engineering, College of Design, Jiaxing University, Jiaxing, China, **3** Department of Marketing, College of Business, Jiaxing University, Jiaxing, China

* 390127406@qq.com

## Abstract

To understand potential users' behavioral intention (BI) to adopt new energy vehicles (NEVs), a media-based perceptions and adoption model (MPAM) of NEVs geared toward potential NEV consumers was constructed based on social cognition theory, the technology acceptance model, the value acceptance model, perceived risk theory, and the MPAM of autonomous vehicles (AVs). A sample survey including 309 NEV potential users was conducted and the results were analyzed through SPSS 24.0 and SmartPLS 3.0 to test the model and verify the research hypotheses. The results show that mass media (MM) has a direct influence on users' social norms (SNs) and part of product perceptions, and an indirect influence on their BI to adopt NEVs; SNs have a direct influence on product perception, and also indirectly affect BI toward NEVs. Product perception directly and significantly affects BI; perceived usefulness (PU), perceived ease of use (PEOU), and perceived enjoyment (PE) have a positive and significant influence on BI, while both perceived cost (PC) and perceived risk (PR) have a negative and significant impact. This study is a theoretical extension of the technology acceptance model (TAM) to green product adoption scenarios, such as NEVs, in the face of the external stimuli from MM information; it proposes product perception variables and media information effects that are different from the MPAM of AVs. The results are expected to greatly promote NEV design and marketing.

## Introduction

New energy vehicles (NEVs) are leading the green transformation and upgrading of the global automotive industry. Major car companies witnessed a robust increase of NEV sales in 2022 due to strong governmental support and the companies' progressive promotion of NEVs. China alone had over 10 million NEVs as of June 2022, accounting for more than half of the NEVs in the world [1]. In 2022, 10.824 million NEVs were sold worldwide according to the "White Paper on the Development of China's New Energy Automobile Industry (2023)," which is a year-on-year increase of 61.6%. Out of this total, 6.884 million were sold in China, accounting for 63.6% of global sales. The main contribution of global automobile sales came from China [2]. Sales growth in Europe and the United States was slower than expected, with year-on-year growth rates of only 16.0% and 52.1%, respectively. As the living standards in

**Data Availability Statement:** All relevant data are within the paper and its Supporting Information files.

**Funding:** This work was supported by Zhejiang Red Culture Research and Inheritance Collaborative Innovation Center [grant number 002CD1902-22-

2-2203]. The funders had no role in the study design, data collection and analysis, decision to publish, or preparation of the manuscript.

**Competing interests:** The authors have declared that no competing interests exist.

China witness constant development with the increase in comprehensive national strength and economy, private cars are becoming prevalent across the country [1]. Petroleum-powered cars, however, create pollution and problems for the environment, including noise, emissions, congestion, and land pollution. The advent of new energy vehicles (NEVs) could reframe these challenges. The NEV industry, a strategic emerging industry in China and a key area supported by the "Made in China 2025" initiative, has witnessed remarkable achievements in recent years under China's policy guidance [1]. China has ranked top for both NEV production and sales for 7 consecutive years since 2015. Taking the NEV industry in China as an example is thus representative.

This study focused on NEV users' adoption actions. Although China enjoys advantages in technological innovation and product development, consumers' acceptance of NEVs is far from satisfactory, given their lack of perception about NEVs' quality, advantages, features, and potential risks. Therefore, market acceptance has not been fully achieved, and there is still much headway to make for potential users to learn and adopt NEVs [3]. Because the application of NEVs is largely determined by user willingness to adopt this green product, we take potential users in the Chinese market as an example to explore the factors and patterns influencing users' NEV adoption behavior, which is of great practical significance for understanding how China can successfully promote the development of NEVs and continue to promote the application of NEVs in the global market in the future.

Various classical models and their extensions have prevailed in studies on NEV adoption behavior, such as the technology acceptance model (TAM) [3–9], the theory of planned behavior (TPB) [10–12], and the unified theory of acceptance and use of technology (UTAUT), along with UTAUT2 [13–15]. However, studies on the public acceptance of driverless technology is richer and offers better possibilities for constructing a theoretical model. Other studies have primarily focused on the adoption of electric and hydrogen vehicles, for which the influence of media information as an external stimulus has been ignored. Zhu et al. (2020) [16] proposed a media-based perceptions and adoption model (MPAM) of autonomous vehicles (AVs) that targets the innovative adoption actions of potential AV users. The MPAM of AVs emphasizes the role of MM. However, as a built-in feature of NEVs, acceptance of autonomous driving among the public or potential users could be replaced by that towards NEVs. Autonomous driving and NEVs are different in nature. Therefore, this study modified the MPAM of AVs proposed by Zhu et al. (2020) [16] and put forward an MPAM of NEVs for potential NEV users in NEV application scenarios. The MPAM of NEVs focuses on potential users' perception of the influence of MM on NEVs. This article discusses the effect of MM on target users' perception and use of NEVs, and employs a dynamic adoption model covering the entirety of the stimulus–perception–action processes. It modifies the MPAM of AVs and puts forward the MPAM of NEVs, which focuses on the influences of social media and social norms (SNs). Meanwhile, factors that may have an effect on the promotion and use of NEVs, such as perceived ease of use (PEOU), perceived enjoyment (PE), and perceived cost (PC) are included, making the product perception variables for NEVs in the model more comprehensive. The MPAM of NEVs offers better explanations for the use of NEVs, so the influencing factors affecting users' willingness to adopt NEVs can be better understood. The model helps to explain the manufacturing and promotion strategies from the perspectives of product design and marketing to support and direct product design and promotion more precisely.

The remainder of this article is structured as follows. The second section introduces the theoretical background and research hypothesis and puts forward the research model; the third section introduces the instrumentation, sample population, sampling technique and data analysis technique; the fourth section describes the research and analysis results; and the fifth

section, the discussion, contains theoretical and practical implications, as well as limitations and future research directions.

## Materials and methods

### Literature review and theoretical framework

Studies related to the acceptance of NEVs include issues such as innovative adoption, innovative diffusion, and green product consumption and public policies. Most studies have been guided by the TPB [10–12], TAM [6–8,17], and innovation diffusion theory [18,19]. More extensively used models and their extensions for innovative adoptions include TAM [3–5], UTAUT, and UTAUT2 [13–15]. TPB includes perceived behavior as a control variable on the basis of the theory of reasoned action (TRA), which provides a more comprehensive attitude behavior theory to explore people's information use [20]. TAM is one of the most influential theoretical extensions of the TRA in confronting new technological environments [21]. It argues that people are influenced by two new determinants—PU and PEOU—in the formation of attitudes towards specific new technologies and the adoption of behaviors. UTAUT integrates eight models of technology adoption, and summarizes the factors affecting user behavior, intention, and actual use behavior into four aspects—performance expectancy, effort expectancy, social influence, and facilitation conditions—and four moderator variables—gender, age, experience, and voluntariness of use. UTAUT2 adds three control variables: hedonic motivation, price value, and habit. Many studies have found that UTAUT and UTAUT2 explain the usage behavior of many scenarios better than the TPB and TAM due to their consideration of many regulatory and control variables [22]. However, UTAUT and UTAUT2 are still considered to be extended TAM models for broader scenarios that extend the external factors and user characteristics of TAM into different new technology backgrounds. As Lim (2018) [21] wrote, the TAM model provides a resilient foundation model for explaining user behavior in today's environment in which new technologies are emerging. In the face of different technical application scenarios, it is only necessary to select and integrate the corresponding extension variables and contextualized behavior-influencing variables based on this basic model to obtain a new model with stronger adaptability to specific technical application scenarios.

Zhu et al. (2020) [16] combined the social cognitive theory (SCT) of Bandura (1986) [23] and the theory of direct perception of Gibson [24] and put forward the MPAM of AVs, which targets the innovative adoption actions of potential users of AVs. Following Gibson's perspective on information, the MPAM of AVs breaks the dynamic process of innovative adoption into three stages: stimulus, perception, and action. Mass media is the environmental information variable in the stimulus stage, and classifies two different types of perceptions—the perception of innovative products and the perception of people (see Fig 1). In the perception stage, users directly derive effective information from the external stimulus, which in turn provides the required information for perception. The stimulus affects the user's decision-making and the generation of innovative acceptance behavior through the perceptual processing of information. Following Bandura's SCT (1986) [23], the MPAM of AVs takes the diverse perceptions of innovative products in TAM, including PU, PEOU, and perceived enjoyment (PE), as the reflection of product information in the perception stage of the three-stage model. In the MPAM of AVs, the variable PU refers to the perception of product factors, while perceived risk (PR) is added to PU to jointly characterize these factors in line with the prospect and value perception theories. According to the MPAM of AVs, people's perception of the external environment does not simply depend on external stimuli: personal as well as others' expectations also affect subsequent innovative acceptance behavior. Therefore, self-efficacy (SE) and

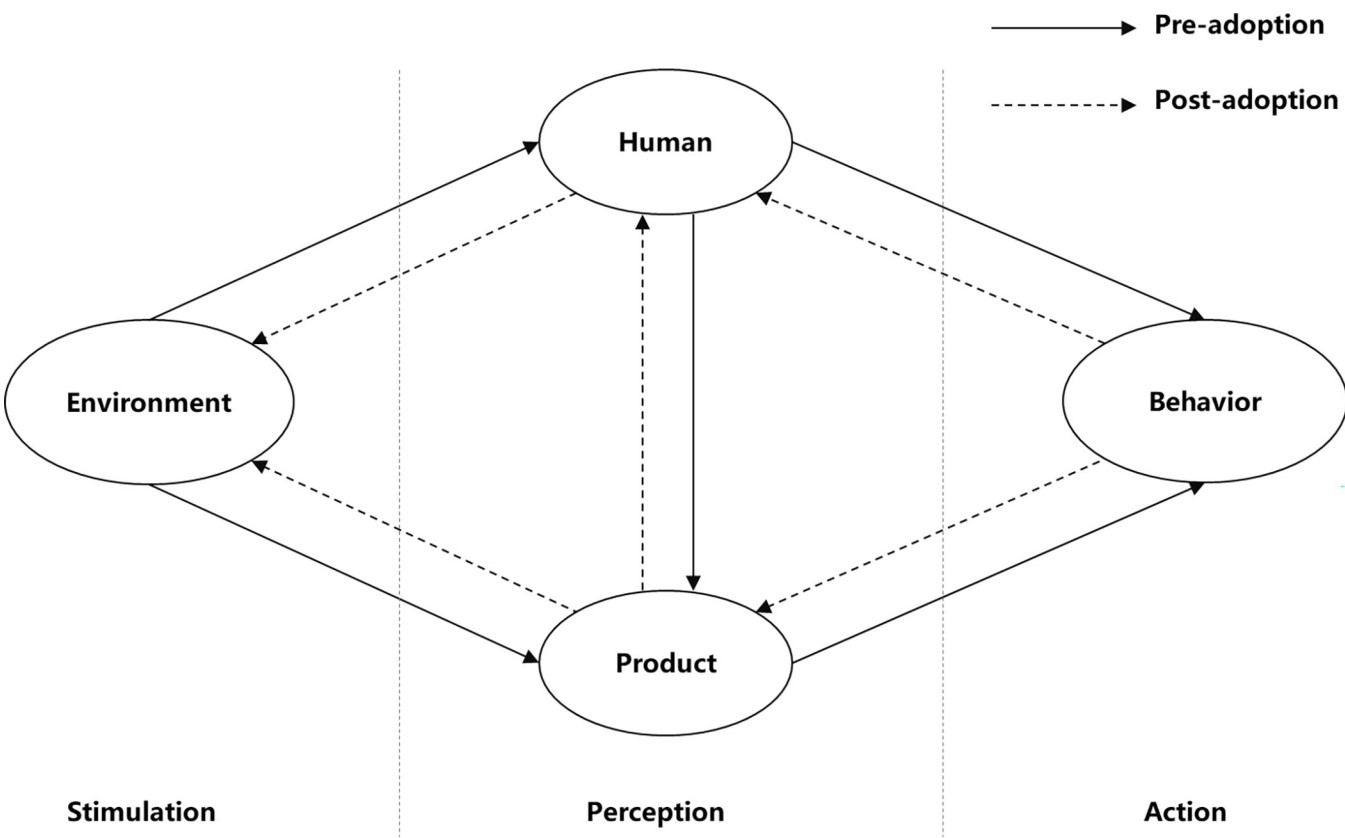

**Fig 1. The three stages of the MPAM of Avs.**

subjective SNs derived from Bandura's social learning theory (SLT) (2009) [25] and Ajzen's TPB (1991) [12] were introduced into the MPAM to represent self-perception and others' expectations.

In contrast to the MPAM of AVs, which highlights the influence of external media represented by MM and social media focuses, this paper focuses on the influence of MM on the willingness of potential users to adopt NEVs. The three-phase framework of MPAM of AVs is also used. The mass media is seen as an environmental stimulus, and the process of its perception is divided into two parts: human perception and product perception. SNs are included in the MPAM of AVs based on SLT (Bandura, 2009) [25], which stands for perceptions of expectation from others. SE is also included based on the TPB [12], which stands for self perception. Compared with AVs, NEVs enjoy advantages in technological maturity and safety. Therefore, SNs are taken as the only variable to represent people's perceptions, and self-perception variables such as SE or perceived behavior control are not included. To measure product perception, all of the variables in the TAM (PU, PEOU, and PE) were included, because NEVs are widely accepted products of emerging technology, so none of the three variables should be omitted. Moreover, PC stemming from the value acceptance model (VAM) by Kim (2010) [26] was also used alongside PR for product perception, because the advantage of NEVs, namely energy-saving and low costs (to which customers are sensitive in purchasing NEVs) are taken into account.

## Model hypotheses

Based on the analysis and modification of the theoretical frameworks mentioned above, an NEV-targeted, media-based user perception and product use model is proposed, as shown in Fig 2.

**Environmental determinants: Mass media.** Mass media information ultimately affects users' willingness to purchase NEVs because reports abound on both the pros and cons of these vehicles. It is assumed that MM influences users' NEV adoption behavior via intermediary factors such as users' beliefs and perceptions [27]. This study suggests that MM affect technology adoption through users' beliefs and perceptions of NEVs. The research results might help NEV manufacturers build an in-depth understanding of how media information influences consumers and then tailor their marketing strategies. Different media channels with various innovation dissemination styles impose their influences in varied ways [28]. MM covers many forms such as television, newspapers, radio, magazines, and the Internet, from which people can obtain considerable amounts of information; this in turn greatly influences people's self-perception, lifestyle, and life attitude. Importantly, MM, as a means of communication, is the major way to develop social perception.

To some extent, the dissemination of information by MM shapes people's perceptions of novel things, which they then relay to change others' mentalities as an intermediary. According to Taylor et al. [29], MM are confirmed to greatly affect SNs via publicity, and some studies have proven that there is a close relationship between MM and users' perceptions of rising technologies [30]. If the advantages of NEVs, such as usefulness, ease of use, enjoyment, low risk, and cost-effectiveness are widely and positively reported by MM, this could significantly and positively influence SNs and users' perception of NEVs, and ultimately affect their adoption behavior. Therefore, the following hypotheses are proposed.

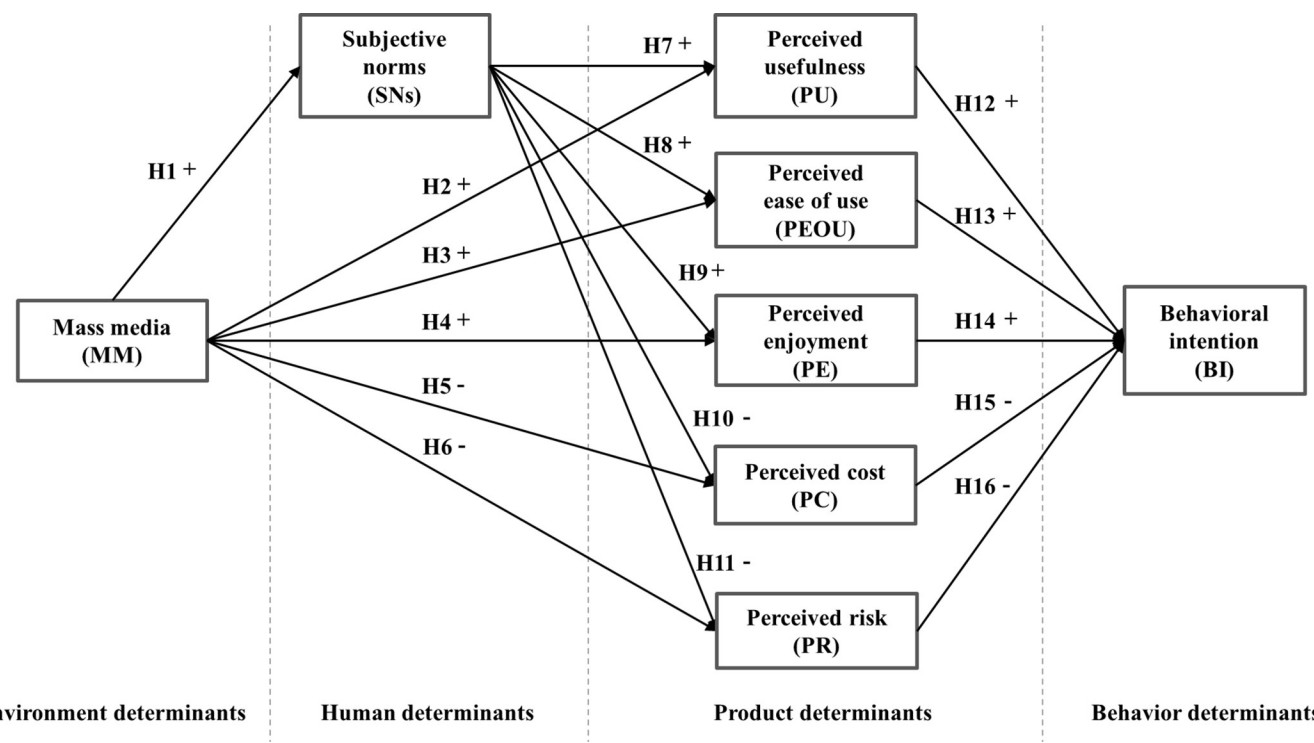

**Fig 2. Media-based perceptions and adoption model of NEVs.**

H1: MM influences SNs in a positive manner.

H2: MM has a positive influence on users' PU of NEVs.

H3: MM has a positive influence on users' PEOU of NEVs.

H4: MM has a positive influence on users' PE of NEVs.

H5: MM has a negative impact on users' PC of NEVs.

H6: MM has a negative impact on users' PR of NEVs.

**Individual determinants: Social norms.**   SCT and diffusion of innovations theory maintain that innovation adoption is primarily the result of a learning or communication process in which SE and SNs are prominent structures that unfold an individual's beliefs about personal abilities and norms. SE refers to confidence in using a technology or product. SNs are the set of perceptions of closely related people or groups that influence individual decision-making. SE and SNs are both forms of self-perception. The former is the spontaneous perception of personal abilities, while the latter involves passive perceptions that reflect what one should do in other people's minds. SNs reveal the fact that our behavior can be largely influenced by other members within a social group. Huang et al. [31] have pointed out that SE has no direct influence on users' NEV inclination because AD rides the future wave, so users are more concerned about the driverless experience rather than the ability to drive themselves. Some researchers have focused on personality differences and pointed out that compared with Westerners, Easterners are more collective and are more likely to be influenced by social groups when making decisions. Because all of the subjects in this study are Chinese, this paper accents the influence of SNs on adoption behavior. A survey in Austin by Bansal et al. [11] revealed that 50% of the population believes that the adoption of NEVs depends on friends and neighbors. SNs are formed by the opinions of people or groups of people closely related to the target user, which can affect personal decision-making [12]. As a form of self-perception, SNs are passive, stemming from perceptions of other entities. Recent studies have confirmed that SNs could have a positive influence on extending the behavioral intention (BI) to engage with NEVs in TAM [6] and UTAUT [13,14]. The hierarchical regression analysis results for the TPB model by Buckley et al. [10] show that SNs are significantly correlated with the BI to use NEVs. Ratten [32] confirmed that before developing a subjective attitude toward new technology, users usually communicate with people close to them, which could help them form new perception judgments. Carolina et al. [33] pointed out that interpersonal influence is closely related to potential users' assessment of the PU of a new technology. The following hypotheses are thus proposed.

H7: SNs have a positive influence on users' PU of NEVs.

H8: SNs have a positive influence on users' PEOU of NEVs.

H9: SNs have a positive influence on users' PE of NEVs.

H10: SNs have a negative impact on users' PC of NEVs.

H11: SNs have a negative impact on users' PR of NEVs.

**Product determinants: PU, PEOU, PE, PR, and PC.**   The TAM model explains users' attitudes toward technology and BI, with PU and PEOU being core variables [34]. PU suggests that users' acceptance is in proportion to their subjective judgment of the usefulness of a new technology or product. In the context of NEVs, PU is defined as the functional benefits of such vehicles to users. Despite the emotional enjoyment users perceive in the meantime, functional usefulness is more important than the emotional, social, or cognitive value in terms of the framework of value theory [7]. According to Zhang et al. [35] PU and PEOU are considered the main factors that may effectively predict users' technology adoption behavior. Panagiotopoulos et al. [6] also noted the significant relationship between these two factors and the user's intention to adopt a product. PC is one of the most frequently cited variables in adoption

behavior studies. PU in this study refers to the user's perceived functional values of NEVs. The following hypothesis is related to PU.

H12: PU positively affects users' willingness to use NEVs.

PEOU refers to the subjective belief that users' acceptance is proportional to the ease of learning a new technology or product, which reflects the operational difficulty of a technology or product, which in turn significantly affects users' recognition of, satisfaction with, loyalty to, and adoption of a product. This is another very important variable in adoption behavior studies, where it is considered as the perceived time and cost of learning something new [36]. PEOU is here defined as the difficulty level for users to learn and use NEVs. Therefore, the following hypothesis is proposed.

H13: PEOU positively affects users' willingness to use NEVs.

PE is the enjoyment and pleasure experienced when using products or services [37]. Motivation is the psychological tendency or internal drive that stimulates and maintains an individual's action and leads to an individual's behavior toward a certain goal. Motivation is an individual internal state, which drives one's behavior to meet the needs of the persona [38]. Hedonic motivation could be interpreted as the individual's behavior in pursuit of happiness and satisfaction [39]. People with high hedonic motivation are more inclined to pursue novel and complex experiences and have stronger risk tolerance than other users [40]: they are thus more willing to accept the novelty and possible risks of NEVs [41]. This is an important predictor for users' acceptance and operations of NEVs [8]. Bay [17] has suggested that the pleasant feelings enjoyed by users while driving NEVs might affect their willingness to buy NEVs. PE in this study is defined as the emotional perception of pleasure, enjoyment, and happiness that users feel when using NEVs. Therefore, the following hypothesis is proposed.

H14: PE positively affects users' willingness to use NEVs.

PC refers to the cost to be paid by users in the process of evaluating, making decisions, and acquiring and using a technology or product [42]. Costs can be related to money, time, psychological or physical effort, and energy, among others. PC in this study primarily refers to monetary cost. Reducing PC is crucial to increase users' perceived value. In the context of NEVs, Xu et al. [43] have shown that PC has a significant impact on users' perceived value. PC in the VAM model also shows a significant impact on users' perceived values in mobile shopping, according to Zhu et al. [44]. In this study, PC is the financial cost to users for adopting an NEV. Therefore, the following hypothesis is proposed.

H15: PC negatively affects users' BI toward NEVs.

PR, a psychological concept, was proposed by Bauer [45] of Harvard University in 1960, referring to negative sentiments among users when they cannot make an accurate judgment about the consequences of purchase behavior [41]. In the context of NEVs, PR is the potential losses users might encounter after their purchase [42]. PR is a major obstacle to user acceptance of NEV. Forsythe et al. [46] have confirmed the relationship between perceived benefits and perceived risks, and the latter may have a direct impact on users' BI. According to Choi and Ji [9], the higher PC, the lower BI is likely to be. Market anxiety increased with the rapid development of NEV, and passive reports kept consumers informed of the disadvantages of NEVs. It is thus necessary to include PR in this study, and the following hypothesis is proposed.

H16: PR has negative impacts on users' BI toward NEVs.

## Methodology

**Instrumentation.** The questionnaire was tailored for NEVs based on previous mature scales. To verify the validity of the contents, a face-to-face pilot survey was conducted, and 30

valid questionnaires were collected. The results were tested for reliability and validity in SPSS 24.0. The subjects claimed that all questionnaire items were easy to understand. Cronbach's α values for each variable were higher than 0.7, indicating that the questionnaire structure met the requirements for reliability. Based on the pilot study, a formal questionnaire covering eight variables with a total of 40 items (MM, SNs, PU, PEOU, PE PR, PC, and BI) was prepared. All of the items were rated on a 5-point Likert scale, ranging from 1 point (strongly disagree) to 5 points (strongly agree). The content of the formal questionnaire is listed in Table 1. The questionnaire also included demographic questions related to basic information such as gender, age, education level, and occupation.

**Sample population and sampling technique.** The questionnaire was primarily distributed to people aged 20–40 years in the Yangtze River Delta region of China, as the core NEV consumers fall into this age bracket. Those aged 20–24 were also included because they are important potential consumers in the future [47]. NEVs are witnessing a robust momentum in their growth worldwide, and China is the world's largest NEV market. A quarter (25.6%) of the newly sold cars in China were NEVs in 2022. Thus, the results of our survey can be used as a Chinese example for the widespread application of this model. Young car designers and car design students constitute a special group with a strong interest in NEVs. They are innovative and progressive, as well as adventurous regarding new technologies and products. They are not only future design decision-makers for NEVs, but also their potential users. It is possible that these representatives can better understand the related product characteristics and user needs, and could also have a deeper understanding of the items in the questionnaire. The studies involving human participants were reviewed and approved by Ethics Committee of Jiaxing University. The participants provided their written informed consent to participate in this study. All participants were adults.

In terms of data collection, the convenience and snowball sampling methods were used. Randomly selected respondents were first interviewed and then asked to recommend respondents with similar characteristics. This method is used for rare groups sampling (car designers are a relatively rare group), which can greatly help reach qualified survey groups, and it is also relatively cheap, so therefore feasible. The questionnaire was made through China's largest online research platform (Sojump) and distributed through China's largest social software platforms (WeChat, QQ); senior automotive design students from many universities in the Yangtze River Delta region of China were invited to participate online. Among the 400 potential respondents, 365 agreed to respond, and further analysis found that 309 were valid responses, as shown in Table 2. The survey included questions related to gender, age, education, and occupation information. Most (88.5%) of the respondents were aged between 20–40, and 90.3% of the respondents had a bachelor's degree or above, in line with the characteristics of the core potential NEV consumers; 55.7% of the respondents were men and 44.4% were women, almost half and half. In terms of sample structure, the respondents are representative.

**Data analysis technique.** In this study, data were collected through an online questionnaire, the model was tested using SmartPLS 3.0, and the partial least squares structure equation model (PLS-SEM) was used to validate the research hypotheses. PLS-SEM was chosen over the covariance-based structural equation method (CB-SEM) for two main reasons. First, compared to theoretical confirmation or comparison (for which CB-SEM is appropriate), this study is exploratory and aims to clarify the prediction of factors influencing user willingness to use NEVs (for which PLS-SEM is more suitable for prediction in this context) [48–50]. Second, in terms of sample distribution, the samples in this study were non-normal, and PLS-SEM has no restrictive assumption about the distribution of data in a multivariate normal distribution compared to CB-SEM [48–50]. Therefore, PLS-SEM estimation was chosen over CB-SEM in this study.

**Table 1. Constructs, items, and their theoretical foundations.**

| Construct/item | Theoretical foundation |
|---|---|
| **Perception Usefulness (PU)** | |
| I think driving an NEV is more efficient and relaxing.<br>I think an NEV offers better driving experience and is more comfortable.<br>I think an NEV makes communication more convenient and smoother.<br>I think an NEV makes car life richer and more exciting.<br>I think an NEV with self-learning ability knows me better and meets my personalized needs. | TAM2 scale (Venkatesh and Davis, 2000) |
| **Perceived ease of use (PEOU)** | |
| I think it's easier to drive an NEV because it does not require frequent checking of the instructions.<br>I think the operation of an NEV is more convenient.<br>I think the operation interface in an NEV is friendly and easy to identify.<br>I think all of the intelligent functions in an NEV are user-friendly and useful.<br>I think it's very convenient to do a remote software update for an NEV. | TAM2 scale (Venkatesh and Davis, 2000) |
| **Perceived risk (PR)** | |
| I think an NEV is prone to spontaneous combustion.<br>I think it's easy for an NEV to leak privacy.<br>I think the autonomous driving of an NEV is prone to error.<br>I think it's easy to decrypt the digital key system of an NEV.<br>I think an NEV is easy for hackers to attack and therefore not safe. | Based on Choi and Ji (2015) |
| **Perceived cost (PC)** | |
| I think the overall cost of purchasing and using an NEV is high.<br>I think charging piles are scarce and the charging is costly and time-consuming.<br>I think it's costly to fix an NEV.<br>I believe that the batteries of NEVs wear out easily, the replacement cost is high, and the car retention rate is low.<br>I think the insurance premiums for an NEV are much higher than for gasoline cars. | Based on Xu et al.(2018) |
| **Perceived enjoyment (PE)** | |
| I think NEVs offer many amusing features like games, videos, and music.<br>I think NEVs have many novel and exciting intelligent features.<br>I think NEVs are interesting for multiple human–machine interactions.<br>I think NEVs have many offline groups and events, which is attractive to me.<br>I think NEVs provide a young, fashionable and cool outlook, which is quite unique. | Based on Van der Heijden (2003,2004) |
| **Mass media (MM)** | |
| Media reports on low carbon and environmental protection affect my BI to accept NEVs.<br>Media reports on young culture affect my BI to accept NEVs.<br>Media reports on smart technology will affect my BI to accept NEVs.<br>Media reports on safety and reliance will affect my BI to accept NEVs.<br>Media reports on customer-centered will affect my BI to accept NEVs. | Based on Taylor and Todd (1995) |
| **Subjective norms (SNs)** | |
| People around me think it's cheap to fuel up an NEV, which is quite suitable for me, as I drive a lot.<br>People around me think an NEV offers a superb driving experience, which is quite suitable for me.<br>People around me think NEVs are smart and convenient, which is quite suitable for me.<br>People around me think NEVs are environment-friendly and emit less carbon, which is quite suitable for me.<br>People around me think NEVs are safe and reliable, so they encourage me to buy. | Based on Taylor and Todd (1995) |
| **Behavioral intention (BI)** | |
| I probably will drive an NEV in the future.<br>I am driving and will continue to drive an NEV.<br>I recommend others drive an NEV.<br>I will switch to an NEV if they are safer and more reliable.<br>I will switch to an NEV if they are smarter and technology-intensive. | TAM2 scale (Venkatesh and Davis, 2000) |

**Table 2. Descriptive statistics.**

| Items | | Total | Percentage |
|---|---|---|---|
| Gender | Male | 172 | 55.7 |
| | Female | 137 | 44.3 |
| Age | Under 30 | 134 | 43.4 |
| | 31–40 | 139 | 45.0 |
| | 41–50 | 25 | 8.4 |
| | Above 50 | 10 | 3.2 |
| Education background | High school graduates or under | 10 | 3.2 |
| | Vocational school graduates | 20 | 6.5 |
| | Undergraduates | 194 | 62.8 |
| | Master students or above | 85 | 27.5 |
| Profession | Company employees | 156 | 50.5 |
| | Students | 38 | 12.3 |
| | Public institutions | 46 | 14.9 |
| | Self-employed | 33 | 10.7 |
| | Civil servants | 12 | 3.9 |
| | Others | 24 | 7.8 |

## Results

### Measurement model assessment

Composite reliability (CR) and Cronbach's α were used to measure the internal consistency of the model, and factor loading and the average variance extracted (AVE) were used to measure the convergent validity of the model. As shown in Table 3, both CR values and Cronbach's α for all of the latent variables in this study exceed 0.7, indicating good internal consistency of the model. The AVE values of all variables exceed 0.5, showing good convergent validity.

Discriminant validity was assessed by comparing the square root of the AVE of each latent variable. Table 4 shows that all square roots of the AVE values exceed the correlation coefficients among the variables related to them, indicating that the model proposed in this study has good discriminant validity.

Henseler pointed out that PLS-SEM could overestimate the factor load while underestimating the correlation between variables—that is, the AVE values would be overestimated. The heterotrait-monotrait ratio (HTMT) should thus be added to the discriminant validity

**Table 3. Test results for all the factors.**

| Latent Variables | No. | Cronbach's $\alpha$ | CR | AVE |
|---|---|---|---|---|
| PU | 5 | 0.896 | 0.923 | 0.706 |
| PEOU | 5 | 0.885 | 0.916 | 0.687 |
| PR | 5 | 0.854 | 0.894 | 0.628 |
| PC | 5 | 0.802 | 0.859 | 0.551 |
| PE | 5 | 0.873 | 0.908 | 0.665 |
| MM | 5 | 0.858 | 0.898 | 0.638 |
| SNs | 5 | 0.884 | 0.915 | 0.684 |
| BI | 5 | 0.860 | 0.900 | 0.642 |

NOTE: PU, Perception usefulness; PEOU, Perceived ease of use; PR, Perceived risk; PC, Perceived cost; PE, Perceived enjoyment; MM, Mass media; SNs, Subjective norms; BI, Behavioral intention.

**Table 4. Test results for discriminant validity.**

| Latent Variables | PU | PEOU | PR | PC | PE | MM | SNs | BI |
|---|---|---|---|---|---|---|---|---|
| PU | **0.840** | | | | | | | |
| PEOU | 0.725 | **0.829** | | | | | | |
| PR | -0.139 | -0.112 | **0.792** | | | | | |
| PC | -0.207 | -0.204 | 0.537 | **0.743** | | | | |
| PE | 0.606 | 0.609 | -0.057 | -0.090 | **0.815** | | | |
| MM | 0.569 | 0.524 | -0.101 | -0.098 | 0.650 | **0.799** | | |
| SNs | 0.583 | 0.602 | -0.176 | -0.275 | 0.619 | 0.660 | **0.827** | |
| BI | 0.541 | 0.535 | -0.246 | -0.303 | 0.454 | 0.556 | 0.737 | **0.801** |

NOTE: The numbers in bold are the square roots of the AVE values for each latent variable. The numbers below are the coefficient relation values among latent variables. PU, Perception usefulness; PEOU, Perceived ease of use; PR, Perceived risk; PC, Perceived cost; PE, Perceived enjoyment; MM, Mass media; SNs, Subjective norms; BI, Behavioral intention.

analysis. The variables have discriminant validity if the HTMT values are less than 0.9. The HTMT values are shown in Table 5, and all are less than 0.9, which indicates that all latent variables in this model have good discriminant validity.

The variance inflation factor (VIF) is an important measure of the severity of multicollinearity in multiple linear regression models. The VIF value of the model in this study is between 1.308 and 2.936, which is less than the critical value of 3.3 (see Table 6), which indicates that there is no multicollinearity in the model in this study, and the model results are relatively stable.

## Structural model assessment

The PLS Algorithm was used to test the suitability of using explanatory variables to predict the result variables, and 5,000 samples were selected with the Bootstrap resampling method for parameter calculation and to evaluate the significance of the model coefficients. The results are shown in Table 7. We found that the variable MM ($\beta = 0.027$, $P > 0.05$) exerts a positive influence on PR, which was inconsistent with the hypothesis, which hints at a negative impact, and the significance evaluation also failed. However, the positive influence of MM on PR is significant, although it is inconsistent with the hypothesis. The remaining variables were all supported at the significance level of 0.05. The influence results are also in line with the hypotheses. More specifically, PU ($\beta = 0.241$, $P < 0.01$), PEOU ($\beta = 0.23$, $P < 0.01$), and PE ($\beta$

**Table 5. Discriminant analysis of HTMT values.**

| Latent Variables | MM | PC | PE | PEOU | PR | PU | SNs | BI |
|---|---|---|---|---|---|---|---|---|
| PU | | | | | | | | |
| PEOU | 0.12 | | | | | | | |
| PR | 0.744 | 0.115 | | | | | | |
| PC | 0.594 | 0.214 | 0.688 | | | | | |
| PE | 0.111 | 0.661 | 0.088 | 0.136 | | | | |
| MM | 0.644 | 0.214 | 0.681 | 0.813 | 0.151 | | | |
| SNs | 0.756 | 0.289 | 0.7 | 0.677 | 0.193 | 0.647 | | |
| BI | 0.647 | 0.328 | 0.514 | 0.604 | 0.262 | 0.606 | 0.842 | |

NOTE: PU, Perception usefulness; PEOU, Perceived ease of use; PR, Perceived risk; PC, Perceived cost; PE, Perceived enjoyment; MM, Mass media; SNs, Subjective norms; BI, Behavioral intention.

**Table 6. VIF values.**

| Latent Variables | VIF | Latent Variables | VIF | Latent Variables | VIF | Latent Variables | VIF | |
|---|---|---|---|---|---|---|---|---|
| MM1 | 1.904 | PE1 | 1.959 | PR1 | 1.614 | SNs1 | 1.712 | |
| MM2 | 2.285 | PE2 | 2.691 | PR2 | 1.638 | SNs2 | 2.858 | |
| MM3 | 2.277 | PE3 | 2.485 | PR3 | 2.007 | SNs3 | 2.851 | |
| MM4 | 1.919 | PE4 | 1.895 | PR4 | 2.417 | SNs4 | 2.172 | |
| MM5 | 1.894 | PE5 | 1.676 | PR5 | 2.633 | SNs5 | 2.018 | |
| PC1 | 1.308 | PEOU1 | 1.896 | PU1 | 2.633 | BI1 | 2.144 | |
| PC2 | 1.507 | PEOU2 | 2.62 | PU2 | 2.74 | BI2 | 2.013 | |
| PC3 | 1.757 | PEOU3 | 2.546 | PU3 | 2.821 | BI3 | 2.645 | |
| PC4 | 2.028 | PEOU4 | 2.465 | PU4 | 2.936 | BI4 | 1.871 | |
| PC5 | 1.539 | PEOU5 | 1.794 | PU5 | 2.076 | BI5 | 1.96 | |

NOTE: PU, Perception usefulness; PEOU, Perceived ease of use; PR, Perceived risk; PC, Perceived cost; PE, Perceived enjoyment; MM, Mass media; SNs, Subjective norms; BI, Behavioral intention.

= 0.149, $P < 0.05$) all have a positive influence on users' BI to adopt NEVs. On the other hand, PC ($\beta = -0.148$, $P < 0.05$). $P < 0.001$) has a negative and significant impact on users' BI, while PR ($\beta = -0.104$, $P = 0.05$) has a negative yet significant impact at critical values. That is, hypotheses H12 through H16 are verified. The variable SNs have a positive and significant influence on PU ($\beta = 0.368$, $P < 0.001$), PEOU ($\beta = 0.454$, $P < 0.001$), PE ($\beta = 0.337$, $P < 0.001$), PC ($\beta = -0.372$, $P < 0.001$), and PR ($\beta = -0.194$, $P < 0.05$), which supports hypotheses H7 through H11. MM also has a positive and significant influence on SNs ($\beta = 0.660$, $P < 0.001$), PU ($\beta = 0.327$, $P < 0.001$), PEOU ($\beta = 0.225$, $P < 0.01$), PE ($\beta = 0.428$, $P < 0.001$), and PC ($\beta = 0.148$, $P < 0.05$), so H1 through H5 are assumed to be true (see Fig 3).

**Table 7. Test results of the model.**

| Hypothesis | Path coefficient | Sample Average | Standard Variance | T Values | P values | Test Results |
|---|---|---|---|---|---|---|
| MM→SNs | 0.66 | 0.662 | 0.038 | 17.463 | 0.000 | True |
| MM→PU | 0.327 | 0.327 | 0.069 | 4.764 | 0.000 | True |
| MM→PEOU | 0.225 | 0.223 | 0.072 | 3.119 | 0.002 | True |
| MM→PE | 0.428 | 0.427 | 0.063 | 6.797 | 0.000 | True |
| MM→PC | 0.148 | 0.152 | 0.075 | 1.976 | 0.048 | Fail |
| MM→PC | 0.027 | 0.03 | 0.082 | 0.333 | 0.739 | Fail |
| SNs→PEOU | 0.368 | 0.369 | 0.061 | 6.000 | 0.000 | True |
| SNs→PEOU | 0.454 | 0.456 | 0.067 | 6.754 | 0.000 | True |
| SNs→PE | 0.337 | 0.338 | 0.065 | 5.205 | 0.000 | True |
| SNs→PC | -0.372 | -0.382 | 0.069 | 5.393 | 0.000 | True |
| SNs→PR | -0.194 | -0.202 | 0.078 | 2.485 | 0.013 | True |
| PU→BI | 0.241 | 0.238 | 0.073 | 3.32 | 0.001 | True |
| PEOU→BI | 0.23 | 0.233 | 0.066 | 3.467 | 0.001 | True |
| PE→BI | 0.149 | 0.149 | 0.067 | 2.222 | 0.026 | True |
| PC→BI | -0.137 | -0.139 | 0.054 | 2.52 | 0.012 | True |
| PR→BI | -0.104 | -0.108 | 0.054 | 1.948 | 0.050 | True |

NOTE: PU, Perception usefulness; PEOU, Perceived ease of use; PR, Perceived risk; PC, Perceived cost; PE, Perceived enjoyment; MM, Mass media; SNs, Subjective norms; BI, Behavioral intention.

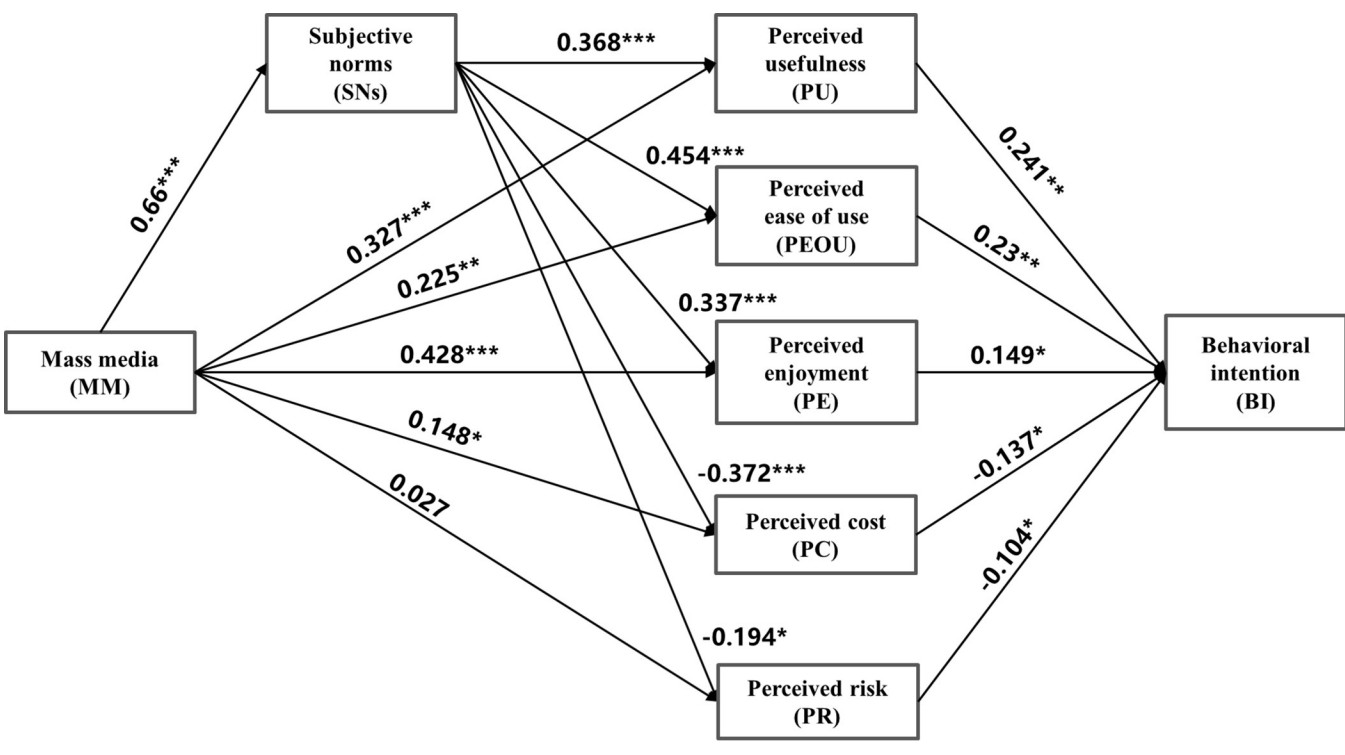

**Fig 3. PLS-SEM results.** NOTE: * means P < 0.05; **means P < 0.01; ***means P < 0.001.

## Discussion

The results show that MM has a significant impact on SNs ($\beta = 0.660$, $P < 0.001$), which indicates that MM has a significant impact on shaping public opinions about NEVs. There is thus a huge development prospect for the low penetration rate of NEVs and the Chinese government's determination to boost the industry. The mainstream MM, a major channel for both the government and businesses to promote NEVs, is thus expected to play a pivotal and positive role. As ever more consumers step up to become NEV drivers, their anticipation for NEV-related information is also expanding. MM reports stimulate market demand and further propel the manufacture of NEVs. Of course, negative reports also plague NEVs, particularly the driverless, AD feature. AD does have many problems, but it is merely one feature of NEVs, not a full excuse for consumers to veto such vehicles. Indeed, consumers are more attracted by advantages such as low cost, decent driving experience, energy-saving, and environmental protection.

This study unveiled that MM exerts a significant influence on three out of five perception variables, namely PU ($\beta = 0.327$, $P < 0.001$), PEOU ($\beta = 0.225$, $P < 0.01$), and PE ($\beta = 0.428$, $P < 0.001$), which is consistent with our hypotheses. PE is influenced the most, followed by PU. Additionally, PE also has a great influence on BI, which means MM reports could strongly reinforce potential users' expectations about the usefulness and enjoyment of NEVs. Relatively speaking, the ease of use and intuitive interfaces of NEV are less attractive than PU and PE, although relevant MM reports still appear to have a positive influence. Considering sales promotion, the wide circulation of MM advertisements or reports is inferior to hands-on experiences such as test rides, as well as to word-of-mouth information exchange with NEV users [51]. This is verified in the present model, as the SN variable exerts the largest influence on PEOU. The data collected show that SNs have significant influences on all of the perception

variables, positive on PU, PEOU, and PE, and negative on PC and PR, which is completely consistent with our model hypotheses. SNs have the greatest impact on PEOU ($\beta = 0.454$, $P < 0.001$), followed by PC ($\beta = 0.372$, $P < 0.001$), PE ($\beta = 0.337$, $P < 0.001$), PU ($\beta = 0.368$, $P < 0.001$), and PR ($\beta = -0.194$, $P < 0.05$). To sum up, MM and its effect on SNs are both important information channels to endow potential users with positive perceptions of NEV.

In terms of PC and PR, the results echo with research hypotheses—that is, SNs have a significant and negative impact on PC and PR, especially on PC, second only to PEOU. Meanwhile, the impact of PC on BI is also very significant ($\beta = -0.137$, $P < 0.001$). PC is more worrying in terms of NEVs, while PR on AD technology is described by MPAM: the lower the cost, the higher users' acceptance and vice versa. At present, NEVs are more expensive in some regards, such as the high cost of battery replacement. At the same time, it is difficult for NEVs to retain value compared to conventional fuel vehicles. Of course, NEVs enjoy obvious cost advantages. For an NEV with a 350 km range, the generally accepted average power consumption is about 15 KWH per 100 km—that is, about 0.15 KWH per kilometer. If it runs 60 km per day, the total cost is about 9 KWH per day—that is, ¥4 every night to fuel it up. In total, the monthly power consumption is about ¥120 [52]. The negative impact of SNs on user willingness to adopt NEVs further reveals that people around actual users find out that the operation cost of NEVs is much lower than that of fossil fuel vehicles based on their experiences, especially for heavy NEV users. At the same time, NEVs also cost lower maintenance fees, as they have no engines. A maintenance operation only requires replacing the brake oil and coolant, while petroleum cars need extra maintenance for the engines, filters, spark plugs, and so on [52]. Moreover, NEVs are equipped with feasible features, such as in-car entertainment, intelligent functions, and a better driving experience. Consumers are willing to pay a premium of less than 10% for NEVs [53].

However, the hypotheses discussing the impact of MM on PR and PC were not verified. The impact of MM on PR ($\beta = 0.027$, $P > 0.05$) was not significant, and while its positive influence on PC ($\beta = 0.148$, $P < 0.05$) just reaches the significant critical value ($P = 0.048$), this was inconsistent with our hypothesis (negative effect) and also inconsistent with MPAM's conclusions. We speculate the reason for this is that MPAM describes AVs and users are strongly concerned about the safety of such vehicles, while NEV technologies are relatively mature, so users' PC is less obvious. However, the influence of SNs on PR and that of PR on BI are both significant. Such results hint that people around users could reduce the PR toward NEVs, thereby increasing the willingness of potential customers. Consumers are quite alert to the risks of NEVs, such as spontaneous combustion, charging difficulty, and the like. These PRs mainly stem from the complaints of nearby users, not from MM. The impact of MM on PC diverges from our hypothesis. We attribute this to the lack of promotions regarding the low costs of NEVs in practice. Much attention has been paid to NEVs' intelligent technology and fashionable and cool appearance, all of which help to create an impression of the expensiveness of NEVs on consumers. It is also true that NEVs are more costly, and media reports elaborate on NEVs' high battery replacement and insurance costs. Alongside these reports, MM's coverage of the features and entertainment of NEVs could increase consumers' PC. In general, MM coverages yield sound effects for the functions and entertainment of NEV while failing to get across the advantages (e.g., ease of operation, low costs, low risk) of NEVs among the public. Conversely, comments on the experience, cost, and risk from communities can generate a positive influence on consumers.

The five perception variables all have a significant impact on BI, with little differences among them, which suggests the balanced consideration among consumers for these five variables. This fully supports the addition of PEOU, PE, and PC to PU, which is in line with Hassenzahl's [54] user experience framework theory, which includes usefulness, ease of use, and

enjoyment. First, the product should be useful to consumers. Second, the product should be easy to use. The design of the product should be tailored to the user's mentality, making it easy to learn and use. Third, interaction with the product should also be emotionally satisfying and pleasant [54]. Products with meaningful features, that are easy to use, and that provide a satisfying emotional experience thus yield a good overall user experience. The results show that PEOU (β = 0.23, $P < 0.01$) has a significant influence on BI. Previous studies also revealed the decisive effect of PEOU and PU on adoption willingness [55]. The results show that PE (β = 0.149, $P < 0.05$) can affect BI, congruent with the results of Hegner et al. [56] and Keszey [39]. Therefore, NEV design should maximize the entertaining features. Previous studies confirmed hedonic motivation to be an important for car adoption [57]. Many people think that cost matters when making a purchase, but would still buy an NEV due to its distinctive recreational features compared with fossil fuel vehicles.

## Theoretical implications

This study explored the factors that influence potential customers' behavioral intention to adopt NEVs by using the MPAM of AVs proposed by Zhu et al. (2020) for driverless vehicles. MPAM is based on TAM, one of the most influential theoretical extensions of TRA. What stands at the core of TAM is that it believes people's PE and PEOU towards new technology are affected by external factors and will affect their adoption willingness and actual behavior at the same time [12,21,58,59]. There have been several versions of TAM over the years, all of which have added to this core. The differences lie in the inclusion and exclusion of attitude or adoption of new technology intention in the model, as well as the expansion of external influencing factors in the face of different new technology backgrounds [22,34,60–62]. As Lim (2018) [21] has argued, TAM provides a resilient foundation for explaining user behavior in today's society, as new technologies are constantly emerging. In the face of different technical application scenarios, it is only necessary to select and integrate the corresponding extension variables and contextualized behavior influence variables based on this basic model to obtain a new model with stronger adaptability to specific technical application scenarios. The TAM-based MPAM of AVs proposed by Zhu et al. (2020) [16] integrates Bandura's (1986) SCT [23] and Gibson's (2014) [24] theory of direct perception. The model breaks the dynamic adoption process into three stages: stimulation, perception, and action. Mass media, which exert a social influence on individual perception, are regarded as external environmental variables in the stimulation stage, and the perception stage is divided into two categories: perception of innovative products and perception of people. Stimulation information from the external environment influences the generation of user decision-making and innovative acceptance behavior willingness through individual perception and processing of information.

This study further strengthens our understanding of users' perception of NEVs during the perception stage based on the MPAM of AVs proposed by Zhu et al. (2020) [16], which targeted user innovation adoption behavior for AVs. The perception of the product in the application scenario of this technology only uses two variables: PU in the TAM and PR in value perception theory. This study focused on users' innovative NEV adoption behavior, which is a more mature and safer scenario in commercialization compared to AVs, and it places more emphasis on user experience. Therefore, we are considering using PU and PEOU in TAM to represent users' basic evaluation of vehicle performance when they are interacting with new technology products, and perceived enjoyment (PE) to represent users' perception and evaluation of the emotional value brought by NEVs, PR to represent users' evaluation of the safety factors of NEVs, and PC to represent users' assessments of the value of NEVs. We agree with the viewpoint of Lim (2018) [21] regarding the TAM model. Therefore, the theoretical

significance of this study lies in the selection and addition of appropriate user perception variables relevant to NEVs, a new technology product, based on the TAM model. This enables the extended model MPAM of NEVs to better explain people's NEV adoption behavior. This study is not a negation of TAM, but rather an extension of TAM to a specific application scenario, which demonstrates its good scalability as a fundamental theory.

## Practical implications

This study provides valuable insights for the marketing and promotion of NEVs: (1) It is better to influence people around users and form SNs, rather than directly using MM when promoting the PEOU of new NEVs, as PEOU is a user experience. (2) It is necessary to guide social public opinion when promoting the PC of NEVs. It could be good to calculate the comprehensive cost of NEVs and compare it with that of traditional fuel vehicles to demonstrate the cost advantages of NEVs. (3) SNs have a greater impact in promoting the PS of NEVs. Therefore, graphic and textual information about the advanced safety technologies, safety certification, and ratings of NEVs can be widely released through social media. At the same time, NEV forums, safety experience activities, interviews with well-known experts, and the like can be held to enhance consumer awareness and confidence in the safety of NEVs. (4) Mass media have a greater influence in promoting PE and PU. Therefore, investment in advertising the recreational function of NEVs should be increased. Private cars are high-end commodities. A car brand can only influence consumer purchasing intentions through long-lasting publicity for a positive image.

This study provides valuable insights for the design of NEVs: (1) It is necessary to strengthen the usability design of NEVs. A user-friendly human-machine interface should be designed, with clear and easily recognizable interface elements, a layout that conforms to existing usage habits, and clear and easy-to-read text content. At the same time, the design should reduce the visual, movement, and psychological loads of users, ensure that they can quickly find the desired content while driving, ensure the shortest possible operation path, and provide encouragement and solutions when problems arise to alleviate anxiety. The principles of consistency, frequency, and importance should also be followed to ensure that the interface color, shape, and font are integrated, with consistent details to ensure a comfortable appearance. The hierarchical order of the interface and where dialogue windows are displayed should be designed according to interaction frequency, and the priority of the control system should be designed according to the importance of user control. (2) The recreational design of NEVs should be enhanced. Designers should create a driving experience that pleases users, and multi-sensory experiences can be integrated in the creation of entertainment scene modes. Machine learning technology can also be used to communicate with users and create emotional partners by studying their behavior patterns. It is also possible to use vehicle networking technology or intelligent transportation systems (among others) to interact in real-time with surrounding environments or vehicles to achieve intelligent emotional interaction. (3) The safety design of NEVs should be improved. It is necessary to improve the safety design of batteries and reduce the potential risk of battery explosion, to improve the stable design that enhances vehicle performance for driver's efficient adaptability to reduce operational accidents, to improve the design of energy management systems to reduce traffic accidents caused by system loss of control, and to improve charging safety design and reduce safety accidents caused by charging. (4) The cost reduction design of NEVs should be strengthened. The design should improve battery energy density, adopt new battery materials, and lift production efficiency while reducing costs. The design should also optimize motor design by reducing the number of motor components and using lightweight materials to reduce costs; a lightweight body design should be used to reduce energy usage costs, and the design should also improve

the charging technology, adopt more efficient chargers, and use intelligent charging technology to reduce costs.

## Limitations and future research directions

TAM is a classic model in the field of innovation adoption and is considered a basic framework for explaining the NEV adoption behavior despite some minor deficits [20]. The results of innovation adoption behavior in this model was measured by the actual adoption behavior in the innovation system. This study was a survey of potential users in China, so the outcome variable of the MPAM of NEVs was a measure of willingness to use NEVs rather than actual use. Although behavior intention can be strongly predictive of actual behavior (according to the TPB), the existence of the intention behavior gap still affects the accuracy and external validity of the conclusions of this study. Future research could conduct large-scale surveys on potential and actual users in a wider range of regions, measuring their willingness and actual use behavior of NEVs, thereby enhancing the universality of the conclusions of this model.

The MPAM of NEVs in this article focused on exploring the impact of MM on the willingness to adopt NEVs. Buying NEVs is green and sustainable consumption, which is encouraged and promoted by both the government and society. Mass media disseminate primarily positive information while private social media give more information. Therefore, compared to the MPAM of AVs, this article only explored the impact of MM on user willingness to adopt NEVs. Correspondingly, the MPAM of NEVs only retained SNs in the variables of the human perception stage, which represent the impact of others' expectations of sustainable consumption behavior on users' perception and willingness to use NEV products. We believe that SE represents users' perception and belief in their ability to use NEVs, and the technical challenge for NEV operation is not high; the impact of MM information dissemination on this is not direct. Future research should consider the impact of more diverse information represented by social media. The impact of user self-awareness based on individual and external control behavior on the perception and willingness to use NEV products should also be included in the perception stage of the model. In the past, research and measurements on perceptual behavior control have emphasized self-perceived control (i.e., SE, perceived control) while neglecting the possible self-perception of externally constrained behavior (e.g., accessibility and availability of products sold by markets). Lim and Weissmann (2023) [20] proposed a theory of behavioral control that divides behavioral control into two categories in terms of concept and operation, covert and overt behavioral control, effectively solving the problem and bridging the intention–behavior gap. These two behavioral control variables could be added to the perception stage of the MPAM of NEVs so the model can better explain the impact and mechanism of external communication information about various NEVs, thus widening the application range of the model.

## Supporting information

**S1 File. Data for CFA.**
(XLSX)

**S2 File. Original and translation of the questionnaire.**
(DOCX)

## Acknowledgments

We thank LetPub (www.letpub.com) for its linguistic assistance during the preparation of this manuscript.

## Author Contributions

**Conceptualization:** Jianjun Pang, Xuan Zhang.

**Data curation:** Jianjun Pang.

**Formal analysis:** Jianjun Pang.

**Funding acquisition:** Jianjun Pang.

**Investigation:** Jianjun Pang.

**Methodology:** Jianjun Pang.

**Project administration:** Jianjun Pang.

**Resources:** Jianjun Pang.

**Software:** Jing Ye.

**Supervision:** Jing Ye, Xuan Zhang.

**Validation:** Jianjun Pang, Jing Ye.

**Visualization:** Jing Ye.

**Writing – original draft:** Jianjun Pang.

**Writing – review & editing:** Jianjun Pang, Xuan Zhang.

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
