## [Decision Letter · Decision Letter 0]

3 Mar 2023

PONE-D-23-01487Factors influencing users’ willingness to use new energy vehiclesPLOS ONE

Dear Dr. Pang,

Thank you for submitting your manuscript to PLOS ONE. After careful consideration, we feel that it has merit but does not fully meet PLOS ONE’s publication criteria as it currently stands. Therefore, we invite you to submit a revised version of the manuscript that addresses the points raised during the review process.

We look forward to receiving your revised manuscript.

Kind regards,

Sudarsan Jayasingh, Ph.D

Academic Editor

PLOS ONE

Journal Requirements:

5. Please ensure that you refer to Figure 3 in your text as, if accepted, production will need this reference to link the reader to the figure.

Additional Editor Comments:

Overall this is well written paper, but the manuscript requires minor revisions. Research methodology need to include the details about sampling techniques, method in which questionnaire was distributed, the year and month it was collected etc. Author need to include a separate section on theoretical and managerial contribution of the research. The conclusion part can be improved by adding the limitation and future research direction.

Reviewers' comments:

Reviewer's Responses to Questions

**Comments to the Author**

1. Is the manuscript technically sound, and do the data support the conclusions?

Reviewer #1: Yes

Reviewer #2: Yes

2. Has the statistical analysis been performed appropriately and rigorously? 

Reviewer #1: Yes

Reviewer #2: Yes

3. Have the authors made all data underlying the findings in their manuscript fully available?

Reviewer #1: Yes

Reviewer #2: Yes

4. Is the manuscript presented in an intelligible fashion and written in standard English?

Reviewer #1: Yes

Reviewer #2: Yes

5. Review Comments to the Author

Reviewer #1: This paper explores the factors that influence users’ willingness to use new energy vehicles. The idea is interesting and the drive toward the adoption of more energy efficient vehicles in light of the climate crisis and the sustainability agenda (which could be acknowledge more prominently in the paper—e.g., see “The sustainability pyramid: A hierarchical approach to greater sustainability and the United Nations Sustainable Development Goals with implications for marketing theory, practice, and public policy” in the “Australasian Marketing Journal”).

The sample size is also adequate and the structural equational modelling technique used to analyze and validate the model is also appropriate. Nonetheless, I do have several suggestions for improvement, which I hope the authors will consider.

First, the paper seems to be positioned based on China. I would encourage the authors to reconsider this and reposition the paper from an international perspective. That is to say, the paper needs to speak more about what resonates internationally, and China could come in as a case of that international issue or agenda.

Second, the TAM is actually a useful theory, but like the authors mentioned, it is inadequate on its own. I reckon it will be good to acknowledge the perspectives of the proponents of TAM and how it should be viewed in order to showcase its true contribution to the advancement of knowledge. For example:

Dialectic antidotes to critics of the technology acceptance model: Conceptual, methodological, and replication treatments for behavioural modelling in technology-mediated environments. Australasian Journal of Information Systems, 22.

Third, one of the issue in the success of green product adoption is the intention-behavior gap, which is influenced by behavioral control. This has not been captured in the present study and thus would make a suitable recommendation for future research to build upon the findings of the present study.

Toward a theory of behavioral control. Journal of Strategic Marketing.

Fourth, the comparison between CB-SEM and PLS-SEM seems broad. Would the present study not meet the assumptions of CB-SEM? I reckon the points used to support PLS-SEM should be chosen strategically based on the nature of the present research itself. Using appropriate references to back up such rationales are also encouraged. Some justifications could be found in the articles below.

Electronic word of mouth on social networking sites: What inspires travelers to engage in opinion seeking, opinion passing, and opinion giving?. Tourism Recreation Research.

How does ethical climate enhance work–family enrichment? Insights from psychological attachment, psychological capital and job autonomy in the restaurant industry. International Journal of Contemporary Hospitality Management.

Fifth, spell out all acronyms under each table so that it is easier for readers to understand.

Finally, it will be good to include dedicated sub-sections on theoretical implications and managerial implications, as well as limitations and future research directions.

I hope these comments will be useful to help the authors improve the quality of the article.

Good luck and all the very best!

Reviewer #2: This is a well-written paper that covers an interesting topic. there are few issues that must be addressed by the authors in order to increase the quality of this paper.

1. In the introduction, it is crucial to emphasize the significance of this work and this can be achieved by highlighting it towards the end. Additionally, it is important to justify the novelty of this paper by outlining its main contributions to the existing literature.

2. It is important to to include a paragraph at the end of the introduction section to show the structure of the paper.

3. Please specify the population of the study and the sampling technique utilized in the methodology and research design section. Additionally, provide a rationale for the selection of your chosen sampling technique and sample size to demonstrate their suitability.

4. Multi-collinearity text should be performed and reported.

5. The discussion section can be improved by comparing the findings with pervious studies.

6. It is recommended to allocate a separate section to highlight both practical and theoretical implications. Please do that after the discussion section.

6. PLOS authors have the option to publish the peer review history of their article (what does this mean?). If published, this will include your full peer review and any attached files.

Reviewer #1: No

Reviewer #2: **Yes: **Ahmad Samed Al-Adwan

---

## [Author Response · Author response to Decision Letter 0]

16 Apr 2023

The formatting of the entire article has been updated to meet PLOS ONE’s style guidelines.

The funding Information is as follows: This work was supported by Zhejiang Red Culture Research and Inheritance Collaborative Innovation Center [grant number 002CD1902-22-2-2203].

The financial Disclosure is as follows: The funders had no role in the study design, data collection and analysis, decision to publish, or preparation of the manuscript.

My ORCID identifier is 0009-0003-0372-8064

The ethics statement has been moved to the Methods section. The content is as follows:

The studies involving human participants were reviewed and approved by Ethics Committee of Jiaxing University. The participants provided their written informed consent to participate in this study. All participants were adults.

5. Please ensure that you refer to Figure 3 in your text as, if accepted, production will need this reference to link the reader to the figure.

We have added the link to Figure 3 in the text.

The references have been updated because the article has been revised. The new reference list is attached as follows. We have also updated the links to specific reference items in the text.

 (1) Attached are the newly added reference items.

[3]Jaiswal D, Kaushal V, Deshmukh AK, et al.What drives electric vehicles in an emerging market?. Marketing Intelligence & Planning.2022;40(6):738-754.

[4]Thilina DK, Gunawardane N. The effect of perceived risk on the purchase intention of electric vehicles: an extension to the technology acceptance model. International Journal of Electric and Hybrid Vehicles. 2019; 11(1): 73-84.

[5]Jaiswal D, Kaushal V, Kant R, et al. Consumer adoption intention for electric vehicles: Insights and evidence from Indian sustainable transportation. Technological Forecasting and Social Change. 2021 Dec10. doi:110.1016/j.techfore.2021.121089. 

[15]Patel H, Shinde Y, Shendge S.Understanding the Adoption and Public Intention to Buy Electric Vehicles in India Using UTAUT. International Journal for Research in Applied Science and Engineering Technology. 2021;9(8): 2528-2535.

[17]Tu JC, Yang C. Key Factors Influencing Consumers’ Purchase of Electric Vehicles. Sustainability.2019;11(14): 3863.

[18]Shanmugavel N, Micheal M. Exploring the Marketing related Stimuli and Personal Innovativeness on the Purchase Intention of E-Vehicles through Technology Acceptance Model. Cleaner Logistics and Supply Chain. 2022 Mar. doi: 10.1016/j.clscn.2022.100029.

[19]Lim WM, Weissmann MA. Toward a theory of behavioral control. Journal of Strategic Marketing. 2023;31(1): 185-211.

[20]Lim WM. Dialectic antidotes to critics of the technology acceptance model: Conceptual, methodological, and replication treatments for behavioural modelling in technology-mediated environments. Australasian Journal of Information Systems. 2018 Jan 22. doi:10.3127/ajis.v22i0.1651.

[21]Venkatesh V, Davis FD. A Theoretical Extension of the Technology Acceptance Model: Four Longitudinal Field Studies. Management Science. 2000;46(2):186-204.

[47]Roland Berger Strategy Consultants. Automobile crowd insight and car purchase decision-making. Automobile & Parts. 2023:4: 46-49.

[48]Hair JF, Ringle CM, Sarstedt M. PLS-SEM: Indeed a silver bullet. Journal of Marketing theory and Practice. 2011; 19(2): 139-152.

[49]Chopra IP, Lim WM, Jain T. Electronic word of mouth on social networking sites: What inspires travelers to engage in opinion seeking, opinion passing, and opinion giving?. Tourism Recreation Research. 2022 Jul4. doi: 10.1080/02508281.2022.2088007.

[50] Lim WM, Cabral C, Malik N, Gupta S. How does ethical climate enhance work–family enrichment? Insights from psychological attachment, psychological capital and job autonomy in the restaurant industry. International Journal of Contemporary Hospitality Management. 2023; 35: 1713-1737.

[51]Yang L,Sun F-L. Research on the Application of Experiential Marketing in the Marketing of New Energy Vehicles. Internal Combustion Engine & Parts. 2022;17: 108-110.

[52]Shi Y-S. Analysis on the use cost of new energy vehicles. Jiangsu Science & Technology Information. 2019;36(17): 45-47.

[58]Ajzen I, Fishbein M. Understanding attitudes and predicting social behavior. Englewood Cliffs. 1980.

[59]Davis FD. A Technology Acceptance Model for Empirically Testing New End-User Information Systems: Theory and Results. Cambridge. Massachusetts Institute of Technology. 1985. Available from: https://core.ac.uk/display/4387241.

[60]Davis FD, Bagozzi RP, Warshaw PR. User Acceptance of Computer Technology: A Comparison of Two Theoretical Models. Management Science.1989; 35(8): 982-1003.

[61]Davis FD. User Acceptance of Information Technology: System Characteristics, User Perceptions and Behavioral Impacts. International Journal of Man-Machine Studies.1993; 38(3):475-487.

 (2) Attached are the deleted reference items.

[1] Yigitcanlar T, Wilson M, Kamruzzaman M. Disruptive impacts of automated driving systems on the built environment and land use: An urban planner’s perspective. Journal of Open Innovation: Technology, Market, and Complexity. 2019; 5(2):24.

[2] Anania EC, Rice S, Walters NW, Pierce M, Winter SR, Milner MN. The effects of positive and negative information on consumers’ willingness to ride in a driverless vehicle. Transport policy. 2018; 72:218–24.

[3] Richard A, Ingrid N, Donna E, Brian S,Susie Q. Effects of Television Modeling on Residential Energy Conservation. Journal of Applied Behavior Analysis.1985; 18(1):18-33

[4] Pakusch C, Stevens G, Boden A, Bossauer P. Unintended effects of autonomous driving: A study on mobility preferences in the future. Sustainability. 2018; 10(7):2404.

[5] Davis FD. Perceived usefulness, perceived ease of use, and user acceptance of information technology. MIS quarterly. 1989:319-40.

[6] Alalwan A A, Baabdullah A M, Rana N P, Tamilmani K, Dwivedi Y K. Examining adoption of mobile internet in Saudi Arabia: Extending TAM with perceived enjoyment, innovativeness and trust. Technol Soc. 2018; 55:100-10.

[7] Gardner B, Abraham C. What drives car use? A grounded theory analysis of commuters’ reasons for driving. Transportation Research Part F: Traffic Psychology and Behaviour. 2007; 10(3):187-200.

[8] Kyriakidis M, Happee R, Winter J C. Public opinion on automated driving: Results of an international questionnaire among 5000 respondents. Transportation research part F: traffic psychology and behaviour. 2015; 32:127-40.

[9] Ro¨del C, Stadler S, Meschtscherjakov A, Tscheligi M, editors. Towards autonomous cars: the effect of autonomy levels on acceptance and user experience. Proceedings of the 6th international conference on automotive user interfaces and interactive vehicular applications; 2014.

[10] Eckoldt K, Knobel M, Hassenzahl M, Schumann J. An experiential perspective on advanced driver assistance systems. It-Information Technology. 2012; 54(4):165-71.

[11] Munir A R, Ilyas G B. Extending the technology acceptance model to predict the acceptance of customer toward mobile banking service in Sulawesi Selatan. International Journal of Economic Research. 2017; 14(4):365-75.

[12] Wang Y, Li Y. Empirical study on consumers' willingness to purchase new energy vehicles based on perceived risk and human involvement. Mathematical statistics and management.2013;32(5):863-872.

[13] Shi H-B, Zou W-N, Xu Y-P. Research on the market of new energy vehicles based on green technology: taking Weihai as an example. Research on science and technology management.2014;34(8):227-232.

[14] Xu G-H, Xu F. Research on the influencing factors of new energy vehicle purchase decision. China's population, resources and environment.2010;20(11):91-95.

[15] Zhang Y-Y, Zhang M-J, Wang Q. Research on the purchase intention of fresh agricultural products under the O2O model based on the perceived income perceived risk framework. China Soft Science.2015;(6):128-138.

[16] Briscoe R, Grush R. Action-based theories of perception. In E.N. Zalta (Ed.), The Stanford encyclopedia of philosophy (Spring 2017 ed.).

[17] Michaels CF. Information, perception, and action: What should ecological psychologists learn from Milner and Goodale (1995)?. Ecological Psychology. 2000;12(3): 241-258.

[18] Mao Z, Lyu J. Why travelers use Airbnb again? An integrative approach to understanding travelers’ repurchase intention. International Journal of Contemporary Hospitality Management.2017;29(9):2464-2482.

[19] Sweeney,Jillian C,Soutar,Geoffrey N. Consumer perceived value:The development of a multiple item scale. Journal of Retailing. 2007;77(2).

[20] Kaur K,Rampersad G.Trust in driverless cars: Investigating key factors influencing the adoption of driverless cars. Journal of Engineering and Technology Management.2018;48: 87-96.

[21] Maslow A H. A theory of human motivation.Psychological Review.1943;50(4):370-396.

[61] Liu P, Yang R, Xu Z. How safe is safe enough for self-driving vehicles? Risk Analysis.2019; 39(2): 315-325.

[22] Awad E,Dsouza S,Kim R,Schulz J,Henrich J,Shariff A. Reply to: life and death decisions of autonomous vehicles. Nature.2020; 579(7797): E3-E5.

[23] Li W, Long R, Chen H, Geng J. A review of factors influencing consumer intentions to adopt battery electric vehicles. Renewable and Sustainable Energy Reviews, 78(April), 318-328.

Comments to the Author

1. the paper seems to be positioned based on China. I would encourage the authors to reconsider this and reposition the paper from an international perspective. That is to say, the paper needs to speak more about what resonates internationally, and China could come in as a case of that international issue or agenda.

Thank you for your comment. In 2022, 10.824 million NEVs were sold worldwide, a year-on-year increase of 61.6%; 6.884 million of these vehicles were sold in China, accounting for 63.6% of global sales. The Chinese market thus takes the largest share in NEV sales. China has ranked top for both NEV production and sales for 7 consecutive years. It can be said that taking NEV industry in China as an example is thus quite representative. We have included related explanations in the Introduction, as well as in the section on sampling. Meanwhile, we believe that more samples from other countries should be included on adoption willingness and actual use behavior to expand the validity of this model. We have explained the limitations and future directions in the Limitations and future research directions section of the Discussion.

2. the TAM is actually a useful theory, but like the authors mentioned, it is inadequate on its own. I reckon it will be good to acknowledge the perspectives of the proponents of TAM and how it should be viewed in order to showcase its true contribution to the advancement of knowledge.

We agree with reviewers’ opinions on TAM. The model used in this article as modified on the basis of the MPAM of AVs, which is an extension of TAM for driverless vehicles. This study is also TAM-based and focused on NEVs. We’ve added appropriate user perception variables to make the model better explain people’s adoption behavior towards NEVs. This article is not a negation of TAM, but rather an extension in a specific application scenario, which can well illustrate the adaptability of the model. We’ve added a sub-section on the theoretical implications after the discussion section.

3. one of the issue in the success of green product adoption is the intention-behavior gap, which is influenced by behavioral control. This has not been captured in the present study and thus would make a suitable recommendation for future research to build upon the findings of the present study.

Thank you for your comment. We have extended the research significance of NEV adoption to issues such as green product adoption and sustainable consumption. Perceived behavior control is an important supplement to the theory of reasoned action (TRA), which is an update to the theory of planned behavior (TPB) of Ajzen (1991). Of course, green product adoption is also related to users’ perceived control behavior. We also agree with the classification and application of covert and overt behavioral control in the theory of behavioral control. We believe that the two control variables could be included in our future study regarding NEV adoption, and we have explained this in detail in the Limitations and future research directions section of the Discussion.

The MPAM of NEVs is modified on the basis of the MPAM of AVs, in which subjective norms and self-efficacy (SE) are used regarding people’s perception. We believe that SE relates to a user’s perceptions and beliefs about his or her ability to successfully adopt NEVs to produce an intended outcome (e.g., ability to manipulate NEVs). Compared to unmanned driving, the handling ability of NEVs is actually far from challenging. At the same time, this study only examined the influence of mass media on NEV users. Because both the government and society support the consumption of NEVs, the media reports regarding NEVs are usually positive, which indicates an indirect impact on SE, example of covert behavioral control, so we did not use it. We entirely agree with the classification and application of theory of behavioral control about the covert and overt behavioral control. We believe that the influence of mass media on overt behavior control variable exists.

4. the comparison between CB-SEM and PLS-SEM seems broad. Would the present study not meet the assumptions of CB-SEM? I reckon the points used to support PLS-SEM should be chosen strategically based on the nature of the present research itself. Using appropriate references to back up such rationales are also encouraged. Some justifications could be found in the articles below.

Thank you for your comment. We chose PLS-SEM over SB-SEM for two reasons. First, this was an exploratory study rather than a theoretical verification or comparison. This study aimed to predict the effects of factors that may influence users’ willingness to adopt NEVs. PLS-SEM is more suitable for this purpose. Second, the samples of this study were non-normal, and PLS-SEM has no restrictive assumption about the distribution of data in a multivariate normal distribution compared to CB-SEM. Therefore, PLS-SEM was chosen over CB-SEM. Relevant reference items have been added.

5. spell out all acronyms under each table so that it is easier for readers to understand.

The acronyms are now spelled out under each table.

6. it will be good to include dedicated sub-sections on theoretical implications and managerial implications, as well as limitations and future research directions.

We have included the theoretical and managerial implications, as well as the limitations and future research directions in the Discussion.

Reviewer #2: This is a well-written paper that covers an interesting topic. there are few issues that must be addressed by the authors in order to increase the quality of this paper.

1. In the introduction, it is crucial to emphasize the significance of this work and this can be achieved by highlighting it towards the end. Additionally, it is important to justify the novelty of this paper by outlining its main contributions to the existing literature.

Thank you for your comment. The model used in this article was modified on the basis of the MPAM of AVs, which emphasized the effect of the mass media as an external stimulus on the adoption behavior of new technology. The MPAM of AVs is an extension of TAM used for driverless vehicles. This study is also TAM-based, but we added appropriate user perception variables concerning NEVs and their application scenarios. In this way, the extended MPAM of NEVs could well explain people’s NEV adoption behavior. This is not a negation of TAM, but rather an extension of it in a specific scenario, which well illustrates its adaptability. We have explained this in detail in the theoretical implications section of the Discussion and have also highlighted it in Introduction. Relevant descriptions could also be found in literature review and theoretical framework parts.

2. It is important to include a paragraph at the end of the introduction section to show the structure of the paper.

Thank you for your comment. We have added such a paragraph at the end of Introduction. The details are as follows:

The remainder of this article is structured as follows. The second section introduces the theoretical background and research hypothesis, and puts forward the research model; the third section introduces the instrumentation, sample population, sampling technique and data analysis technique; the fourth section describes the research and analysis results; and the fifth section, the discussion, contains theoretical and practical implications, as well as limitations and future research directions.

3. Please specify the population of the study and the sampling technique utilized in the methodology and research design section. Additionally, provide a rationale for the selection of your chosen sampling technique and sample size to demonstrate their suitability.

Thank you for your comment. We have added related content about the sample population and sampling technique. Attached are the details:

The questionnaire was primarily distributed to people aged 20–40 years in the Yangtze River Delta region of China, as the core NEV consumers fall into this age bracket. Those aged 20–24 were also included because they are important potential consumers in the future [47]. NEVs are witnessing a robust momentum in their growth worldwide, and China is the world’s largest NEV market. A quarter (25.6%) of the newly sold cars in China were NEVs in 2022. Thus, the results of our survey can be used as a Chinese example for the widespread application of this model. Young car designers and car design students constitute a special group with a strong interest in NEVs. They are innovative and progressive, as well as adventurous regarding new technologies and products. They are not only future design decision-makers for NEVs, but also their potential users. It is possible that these representatives can better understand the related product characteristics and user needs, and could also have a deeper understanding of the items in the questionnaire. The studies involving human participants were reviewed and approved by Ethics Committee of Jiaxing University. The participants provided their written informed consent to participate in this study. All participants were adults.

In terms of data collection, the convenience and snowball sampling methods were used. Randomly selected respondents were first interviewed and then asked to recommend respondents with similar characteristics. This method is used for rare groups sampling (car designers are a relatively rare group), which can greatly help reach qualified survey groups, and it is also relatively cheap, so therefore feasible. The questionnaire was made through China’s largest online research platform (Sojump) and distributed through China’s largest social software platforms (WeChat, QQ); senior automotive design students from many universities in the Yangtze River Delta region of China were invited to participate online. Among the 400 potential respondents, 365 agreed to respond, and further analysis found that 309 were valid responses, as shown in Table 2. The survey included questions related to gender, age, education, and occupation information. Most (88.5%) of the respondents were aged between 20–40, and 90.3% of the respondents had a bachelor’s degree or above, in line with the characteristics of the core potential NEV consumers; 55.7% of the respondents were men and 44.4% were women, almost half and half. In terms of sample structure, the respondents are representative.

4. Multi-collinearity text should be performed and reported.

We have added explanations about multicollinearity and a table of VIF values. The VIF values of the model used in this article fell between 1.308 to 2.936, which were all less than the critical value of 3.3.

5. The discussion section can be improved by comparing the findings with previous studies.

This article offers a theoretical extension of TAM to NEV adoption, which is a green product adoption scenario. It puts forward product perception variables and media information effects that are different from the MPAM of AVs. It also provides practical insights for the product design and promotion of NEVs. We have added the theoretical implications to the Discussion and compared them with those of previous studies.

6. It is recommended to allocate a separate section to highlight both practical and theoretical implications. Please do that after the discussion section.

We have included the theoretical and practical implications, as well as limitations and future research directions, in the Discussion.

---

## [Editor Report · Decision Letter 1]

2 May 2023

Factors influencing users’ willingness to use new energy vehicles

PONE-D-23-01487R1

Dear Dr. Pang,

We’re pleased to inform you that your manuscript has been judged scientifically suitable for publication and will be formally accepted for publication once it meets all outstanding technical requirements.

Kind regards,

Sudarsan Jayasingh, Ph.D

Academic Editor

PLOS ONE
---

## [Editor Report · Acceptance letter]

10 May 2023

PONE-D-23-01487R1 

Factors influencing users’ willingness to use new energy vehicles 

Dear Dr. Pang:

I'm pleased to inform you that your manuscript has been deemed suitable for publication in PLOS ONE. Congratulations! Your manuscript is now with our production department. 

Kind regards, 

on behalf of

Dr. Sudarsan Jayasingh 

Academic Editor

PLOS ONE